# Occupational exposures and mitigation strategies among homeless shelter workers at risk of COVID-19

Carol Y. Rao[1]*, Tashina Robinson[1], Karin Huster[2], Rebecca L. Laws[1], Ryan Keating[1], Farrell A. Tobolowsky[1,3], Temet M. McMichael[2,3], Elysia Gonzales[2], Emily Mosites[1]

**1** COVID-19 Response Team, Centers for Disease Control and Prevention, Atlanta, Georgia, United States of America, **2** Public Health-Seattle & King County, Seattle, Washington, United States of America, **3** Epidemic Intelligence Service, Centers for Disease Control and Prevention, Atlanta, Georgia, United States of America

* CRao@cdc.gov

**Data Availability Statement:** All relevant data are within the manuscript and its Supporting information files.

## Abstract

### Objective

To describe the work environment and COVID-19 mitigation measures for homeless shelter workers and assess occupational risk factors for COVID-19.

### Methods

Between June 9-August 10, 2020, we conducted a self-administered survey among homeless shelter workers in Washington, Massachusetts, Utah, Maryland, and Georgia. We calculated frequencies for work environment, personal protective equipment use, and SARS-CoV-2 testing history. We used generalized linear models to produce unadjusted prevalence ratios (PR) to assess risk factors for SARS-CoV-2 infection.

### Results

Of the 106 respondents, 43.4% reported frequent close contact with clients; 75% were worried about work-related SARS-CoV-2 infections; 15% reported testing positive. Close contact with clients was associated with testing positive for SARS-CoV-2 (PR 3.97, 95%CI 1.06, 14.93).

### Conclusions

Homeless shelter workers may be at risk of being exposed to individuals with COVID-19 during the course of their work. Frequent close contact with clients was associated with SARS-CoV-2 infection. Protecting these critical essential workers by implementing mitigation measures and prioritizing for COVID-19 vaccination is imperative during the pandemic.

**Funding:** The authors received no specific funding for this work.

**Competing interests:** The authors have declared that no competing interests exist.

## Introduction

Severe Acute Respiratory Syndrome Coronavirus-2 (SARS-CoV-2), the virus that causes coronavirus disease 2019 (COVID-19), has spread rapidly among people experiencing homelessness in some homeless shelters throughout the United States [1,2]. In several reported outbreaks, homeless shelter workers have also been infected with the virus [3,4]. Homeless shelter workers provide critical infrastructure services [5] and often work in shared spaces with the potential for prolonged close contact with other staff and clients. A meta-analysis of COVID-19 in homeless shelters estimated a pooled prevalence among shelter workers of 14.8% during outbreaks and 1.55% during non-outbreak situations [6]. Homeless shelter workers may be at increased risk for COVID-19 due to frequent exposure to disease and infectious agents in the workplace.

Homeless shelters provide an essential service and, like many congregate settings, have remained open during the COVID-19 pandemic. Homeless shelters have previously experienced outbreaks of tuberculosis, hepatitis A, and invasive bacterial disease [7–10]. However, very little information exists about the risks of exposure for homeless shelter workers. In a study of occupational exposures to infectious agents, an estimated 32.4% of community and social services sector workers are exposed > 1 time/month to infection or disease; 7.7% are estimated to be exposed > 1 time/week [11]. Occupational health protections for shelter workers include following infection control processes, bringing their adult immunization status up to date, encouraging annual influenza and Hepatitis B vaccination, and screening/testing for tuberculosis [12,13].

Other crowded occupational environments, including long-term care facilities, correctional facilities, military facilities and cruise ships, have been associated with high risk of exposures and transmission of COVID-19 among workers [14–19]. Although various mitigation measures have been recommended to reduce homeless shelter worker COVID-19 risks [20], the implementation of these measures is unknown. To our knowledge, risk factors for COVID-19 among homeless shelter workers have not been previously described. We conducted a multi-site cross-sectional survey of homeless shelter workers to better understand SARS-CoV-2 occupational exposures, job practices, and COVID-19 mitigation measures.

## Materials and methods

### Homeless shelter selection and recruitment

Local public health and healthcare collaborators (e.g., local health departments, nongovernmental organizations, local government agencies) in Seattle, Washington; Boston, Massachusetts; Salt Lake City, Utah; Baltimore, Maryland; and across Georgia identified homeless shelters in their jurisdictions where at least one staff member had tested positive for SARS-CoV-2. We informed the shelter administrators of the survey objectives and requested participation of their staff. All staff who work at a shelter facility were eligible to participate. Public health partners or shelter administrators sent a recruitment email with a link to the online survey to all workers. The online survey was open between June 9 and August 10, 2020. Participation was voluntary and anonymous. At least two follow-up emails were sent to encourage participation.

### Survey administration

The Centers for Disease Control and Prevention (CDC) and Public Health—Seattle & King County (PHSKC) developed the standardized online survey (S1 File) that included questions on demographics, work environment, possible SARS-CoV-2 exposures, and workplace

COVID-19 mitigation strategies such as availability and use of personal protective equipment (PPE), hand hygiene facilities, and masks. The survey included questions about SARS-CoV-2 testing history, including number of testing events, test results, testing facility, symptoms around the time of testing, job practice while symptomatic, and medical services seeking behavior. We asked whether the test was a blood test, assuming that participants may not understand the terminology for antibody testing (i.e., blood test) versus molecular testing (i.e., nasopharyngeal swab). Study data were collected and managed using REDCap electronic data capture tools hosted at CDC.

### Data analysis

We conducted the analyses using Stata/SE 16.0 and R version 4.0.2. Body Mass Index (BMI) was calculated by multiplying weight in pounds by a conversion factor of 703 and dividing by height in inches squared. Job titles and job descriptions were used to categorize job positions as primarily administrative (e.g., supervisors, office administration, information technology, accounting) or client engagement (e.g., case manager, food server, floor monitor, housing advocate, social worker). Individual shelters that were managed by the same organization were grouped into shelter networks for analyses. Because some workers were tested more than once, a worker was classified as positive for SARS-CoV-2 if he/she self-reported a positive non-blood test result for at least one testing event. Workers who reported "Don't know" for a test result were categorized as a non-positive. We calculated frequencies, medians, and ranges to describe demographic and work characteristics of participants. To explore associations between SARS-CoV-2 positivity and participant characteristics, we used generalized linear models with a binary outcome (COVID-19 positivity according to at least one non-blood test) and a log link, clustered by shelter network (to allow for intragroup correlation), to produce unadjusted prevalence ratios (PR) and 95% confidence intervals (CI). Frequency of close contact, defined as less than 6 feet for more than 15 minutes at a time, was dichotomized into low (never, rarely, a few times a month) and high (a few times a week to a few times a day). We produced PRs comparing workers who reported testing positive at least once to workers who reported testing negative for every test by demographics (e.g., age, sex, ethnicity), work environment (e.g., hours worked, frequency of close contact, COVID-19 mitigation strategies implemented by facility), and attitudes (e.g., thought about quitting).

This activity underwent human subjects ethics review by CDC and was conducted consistent with applicable federal law and CDC policy (45 C.F.R. part 46, 21 C.F.R. part 56; 42 U.S.C. §241(d); 5 U.S.C. §552a; 44 U.S.C. §3501 et seq). Completing the survey was voluntary.

## Results

Among 17 shelter networks that represented 27 individual shelters, 106 homeless shelter workers (range per shelter network, 1–33) completed the online survey. The median age of participants was 42 years (range 21–67); 65 (61%) were female and 55 (52%) were non-Hispanic White (Table 1). Of the 106 participants, 23 (22%) were current smokers/vapers, 15 (14%) reported having a chronic lung disease, and median BMI was 28.2 (range 18.1–58.1). The median number of people living in workers' households was 3 persons (range 1–9 persons). Participants reported working at the shelter for a median of 40 hours per week (range 1–60 hours) and for a median of 20 months (range 0–336 months); 41 workers (39%) worked at their shelter for ≤12 months with 20 (19%) for ≤6 months.

Sixty-three participants (59%) reported some sort of client engagement as part of their regular work duties, including case management, providing medical and mental health care, client intake, client outreach, client screening, serving food to clients, providing education and

**Table 1. Demographics, job practices and COVID-19 mitigation measures of homeless shelter workers (N = 106).**

| Demographics | Number (%[+]) | Job practices, mitigation measure and attitudes | Number (%) |
|---|---|---|---|
| **Geographic location** | | **Any mask use at work[#]** | |
| GA | 16 (15.1) | Most/all of the time | 91 (85.8) |
| Boston, MA | 33 (31.1) | Sometimes | 11 (10.4) |
| Baltimore, MD | 2 (1.9) | Rarely/never | 1 (0.9) |
| Salt Lake City, UT | 25 (23.6) | **Frequency of close contact with clients[**]** | |
| Seattle, WA | 30 (28.3) | A few times a day | 34 (32.1) |
| **Age** | | A few times a week | 12 (11.3) |
| 21–30 years old | 23 (21.7) | A few times a month | 10 (9.4) |
| 31–40 years old | 28 (26.4) | Rarely | 2 (1.9) |
| 41–50 years old | 24 (22.6) | Never | 46 (43.4) |
| >50 years old | 31 (29.2) | **Frequency of direct physical contact with clients[***]** | |
| **Sex** | | A few times a day | 12 (11.3) |
| Male | 35 (33.0) | A few times a week | 10 (9.4) |
| Female | 65 (61.3) | A few times a month | 4 (3.8) |
| **Gender identity** | | Rarely | 13 (12.3) |
| Male | 38 (35.8) | Never | 65 (61.3) |
| Female | 64 (60.4) | **Frequency of close contact with coworkers[**]** | |
| Non-Binary | 2 (1.9) | A few times a day | 61 (57.5) |
| **Race/Ethnicity** | | A few times a week | 12 (11.3) |
| Non-Hispanic White | 55 (51.8) | A few times a month | 11 (10.4) |
| Non-Hispanic Black | 26 (24.5) | Rarely | 15 (14.2) |
| Non-Hispanic Other | 6 (5.7) | Never | 7 (6.6) |
| Hispanic | 15 (14.2) | **Cleaning activities as part of normal job** | |
| **BMI ≥ 30** | | Yes | 60 (56.6) |
| Yes | 39 (36.8) | If yes, trained on cleaning for SARS-CoV-2 | 34/60 (56.7) |
| No | 56 (52.8) | No | 43 (41.8) |
| **Smoking** | | **Mitigation measure implemented by facility** | |
| Current smoker/vaper | 23 (21.7) | Increased handwashing | 88 (83.0) |
| Past smoker | 22 (20.8) | Safe distancing (≥6 ft) | 82 (77.4) |
| **Received flu vaccine this year** | | Masks for staff/clients | 97 (91.5) |
| Yes | 67 (63.2) | Provision of PPE for staff | 80 (75.5) |
| No | 38 (35.9) | No measures implemented | 1 (0.9) |
| **Any underlying conditions[^]** | | **Agree with organization's response to COVID-19** | |
| Yes | 30 (28.3) | Yes | 67 (63.2) |
| No | 71 (67.0) | No | 14 (13.2) |
| **Chronic lung disease** | | Don't Know | 12 (11.3) |
| Yes | 15 (14.2) | **Worried about being infected with SARS-CoV-2 due to job** | |
| No | 86 (81.1) | Yes | 80 (75.4) |
| **Has paid sick leave** | | If yes, thought about quitting due to COVID-19 | 19/80 (23.8) |
| Yes | 96 (90.6) | No | 21 (19.8) |
| No | 6 (5.7) | **Family supportive of worker's job** | |
| **Primary job duties** | | Yes | 87 (82.1) |
| Administrative | 43 (40.6) | No | 9 (8.5) |
| Client engagement | 63 (59.4) | **Family worried about worker being infected due to job** | |
| **Has formal health education[*]** | | Yes | 83 (78.3) |
| Yes | 30 (28.3) | No | 19 (17.9) |
| No | 73 (68.9) | **Worker worried about family being infected due to worker's job** | |

*(Continued)*

**Table 1.** (Continued)

| Demographics | Number (%[+]) | Job practices, mitigation measure and attitudes | Number (%) |
|---|---|---|---|
| **Have another job** | | Yes | 66 (62.3) |
| Yes | 18 (17.0) | No | 38 (35.9) |
| No | 86 (81.1) | | |
| **Length of employment** | | | |
| 0–6 months | 20 (18.9) | | |
| 7–12 months | 21 (19.8) | | |
| 13–36 months | 25 (23.5) | | |
| 37–60 months | 14 (13.2) | | |
| >60 months | 23 (21.7) | | |

[+]May not sum to 100% in some categories due to missing data.

*For example, nursing, medicine, or emergency medical technician.

**Close contact = within 6 feet for ≥15 minutes.

***Direct physical contact = touching.

^Chronic lung disease, High blood pressure, chronic kidney or liver disease, diabetes mellitus, rheumatoid arthritis, heart disease.

#Disposable or reusable mask.

employment advice, monitoring clients while at the shelter, and janitorial activities (Table 1). Almost one third of participants reported close contact with clients a few times per day (34 participants; 32%); 65 participants (61%) reported that they never have direct physical contact (i.e., touching) with clients. Of the 43 participants categorized as administrative job duties, 22 (51.2%) reported close contact or direct physical contact with clients. Over half of participants reported close contact with coworkers a few times per day (61 participants; 58%). Many homeless shelter workers reported that cleaning was part of their normal duties (60 participants; 57%); of those, 34 (57%) reported receiving training on how to clean an area after a client with known COVID-19 leaves the shelter. Most workers (75%) reported being worried about becoming infected due to their jobs, and of those, 24% had thought about quitting. Although their families were supportive of their jobs, their families also worried about workers being infected due to their jobs (Table 1).

All 17 shelter networks had implemented at least one COVID-19 prevention measure. At the participant level, the most common mitigation measures were use of masks by staff or clients (97/106, 92%) and increased handwashing (88/106, 83%) (Table 1). Eighty-six percent of staff (n = 91) reported wearing a mask at least most of the time while at work. Among those who reported close contact with clients (n = 58), 47 (81%) reported wearing a mask most or all of the time. Among those who reported close contact with coworkers (n = 99), 62 (77%) reported wearing masks when in close contact with coworkers. Among those who reported having direct physical contact with clients (n = 39), 22 reported wearing gloves (56%). Among those who reported having close contact with a person with known COVID-19 (n = 38); 29 (76%) reported wearing a disposable mask most or all of the time, and 24 (63%) reported wearing gloves (Table 2). All believed that their close contact to a person with known COVID-19 occurred at work while 2 workers (5%) also said they had contact at home.

Of the 106 participants, 77 reported being tested for SARS-CoV-2 with 62% of participants (48/77) reported undergoing testing for SARS-CoV-2 more than once. For the 77 participants who reported being tested at least once, the median number of testing events per participant was 2 (range: 1–10). Of the 187 testing events reported by the 77 participants, 9 were blood tests (assumed to be serological testing), 170 were non-blood tests (assumed to be molecular

**Table 2. Homeless shelter worker high risk activities and use of masks and personal protective equipment (PPE) during encounters (N = 106).**

| Characteristic | Number (Percent) |
|---|---|
| **Has close* or direct physical** contact with clients** | **58 (54.7%)** |
| Worker used mask# most/all of the time when in close contact* with clients | 47 (81.0%) |
| Worker used mask# sometimes when in close contact* with clients | 9 (15.5%) |
| Worker used mask# rarely/never when in close contact* with clients | 2 (3.5%) |
| Clients used mask# most/all of time during close contact* | 25 (43.1%) |
| Clients used mask# sometimes during close contact* | 20 (34.5%) |
| Clients used mask# rarely/never during close contact* | 13 (22.4%) |
| **Has direct physical** contact with clients** | **39 (67.2%^)** |
| Used gloves most/all of the time when in direct physical contact** with clients | 22 (56.4%) |
| Used gloves sometimes when in direct physical contact** with clients | 7 (17.9%) |
| Used gloves rarely/never when in direct physical contact** with clients | 9 (23.1%) |
| **Touch clients' belongings or shared items** | **56 (52.8%)** |
| Used gloves when touching clients' belongings/shared items | 48 (85.7%) |
| **Has close contact* with coworkers** | **99 (93.4%)** |
| Used mask# when in close contact* with coworkers | 62 (76.5%) |
| **Had close contact* with person with known COVID-19** | **38 (35.8%)** |
| Used mask most/all of the time when interacting with known COVID-19 | 29 (76.4%) |
| Used mask sometimes when interacting with known COVID-19 | 3 (7.9%) |
| Used mask rarely/never when interacting with known COVID-19 | 3 (7.9%) |
| Used gloves when interacting with known COVID-19 | 24 (63.2%) |
| Used gown when interacting with known COVID-19 | 2 (5.3%) |
| Used respirator (N95) when interacting with known COVID-19 | 8 (21.1%) |
| Used goggles when interacting with known COVID-19 | 1 (2.6%) |
| Did not use masks or any PPE when interacting with known COVID-19 | 2 (7.1%) |

^Denominator = 58 (workers who had close or direct physical contact with clients).

#Disposable or reusable mask.

*Close contact = within 6 feet for ≥15 minutes.

**Direct physical contact = touching.

testing), and 8 did not select a test type (Table 3). The average number of days for a participant to receive their test result was 3.1 days (range 0–10 days). Participants reported working while waiting for test results for 118 testing events (63%). Of the 187 tests, 114 testing events (61%) occurred at the workplace. Among 21 positive non-blood tests (4 workers tested positive multiple times), 16 (76%) were from participants who were symptomatic around the time of the test. There were 15 tests among 10 people who indicated "Don't know" and/or did not answer for a test result, who were categorized as a non-positive. One worker who tested positive worked 1 day while symptomatic and while waiting for their result (Table 3). The worker initially had a sore throat and headache with additional symptoms after receiving the positive test result. Overall, 16 workers reported a positive test result with 46.7% reporting primarily administrative job duties and 53.3% reporting job duties with client engagement; 2 reported a positive blood test and 15 reported a positive non-blood test (one participant tested positive by both blood and non-blood test), for an overall prevalence rate of 15% (16/106). Of the 14 participants who answered the question about where they thought they were infected, 12 participants who tested positive believed they were infected at work.

**Table 3. Homeless shelter workers' self-reported COVID-19 testing type, test locations, and symptoms and mitigation measures around the time of testing among.**

| Test event characteristics^ | Number (%*) |
|---|---|
| **Test type**** | |
| Non-blood test | 170/187 (90.9) |
| Blood test | 9/187 (4.8) |
| **Worked while waiting for test results** (missing n = 4) | 118/187 (63.1) |
| **Testing location** | |
| Workplace | 114/187 (61.0) |
| *Worked while waiting for result* | *97/118 (85.1)* |
| Medical provider | 45/187 (24.1) |
| *Worked while waiting for result* | *16/45 (35.6)* |
| Other | 25/187 (13.4) |
| *Worked while waiting for result* | *5/25 (20.0)* |
| **Days to get test results** | |
| <3 days | 108/187 (57.8) |
| 3–7 days | 62/187 (33.2) |
| +7 days | 9/187 (4.8) |
| **Symptomatic 1 month or 2 months after testing event** | 58/187 (31.0) |
| **Among positive non-blood testing events^^** | **Number (%*)** |
| **Symptomatic (n = 21 testing events)** | 16/21 (76.2) |
| Worked while having symptoms | 1/16 (6.30) |
| Sought medical care for symptoms | 10/16 (62.5) |
| **Delivery of positive test result (n = 15 participants)** | |
| Supervisor | 3/15 (20.0) |
| Called testing provider | 1/15 (6.70) |
| Health department staff | 6/15 (40.0) |
| Primary care physician | 1/15 (6.70) |
| Other | 2/15 (13.3) |
| **Management instructions after COVID-19 diagnosis (n = 15 participants)** | |
| Stay home and isolate | 12/15 (80.0) |
| Continue to work | 0/15 (0.0) |
| No instruction provided | 2/15 (13.3) |
| **Where they thought they were infected (n = 15 participants)** | |
| Home or in Community | 0/15 (0.0) |
| Work | 12/15 (80.0) |
| Don't know where | 2/15 (13.3) |

^ N = 187 tests from 77 participants.

^^ n = 21 tests from 15 participants.

*Some categories may not sum to 100% due to missing data.

**Non-blood tests were assumed to be molecular testing while blood tests were assumed to be serological tests.

When analyzing the 77 participants who reported at least one testing event, we identified demographic characteristics that were associated with testing positive for SARS-CoV-2, including having a BMI ≥30 (PR 1.86, 95% CI 1.31, 2.63; p = 0.001) or identifying as Non-Hispanic Black race (PR 2.00, 95% CI 1.23, 3.26; p = 0.01)(Table 4). Reporting frequent close contact with clients (PR 3.97, 95% CI 1.06,14.93; p = 0.04) or using gloves when interacting with a person with known COVID-19 (PR 3.90, 95% CI 1.36, 11.19; p = 0.01) was associated with

**Table 4. Unadjusted prevalence ratios (PR) and 95% confidence intervals (CI) for factors associated with COVID-19 positivity^ among homeless shelter workers (N = 77 participants who reported being tested).**

| Characteristic | PR (95% CI) | p-value[+] |
|---|---|---|
| Age (>40 years old) | 1.54 (0.85, 2.78) | 0.15 |
| Sex (Female) | 1.63 (0.32, 8.28) | 0.55 |
| Hispanic ethnicity | 2.67 (0.60, 11.9) | 0.20 |
| **Non-Hispanic Black race** | **2.00 (1.23, 3.26)** | **0.01** |
| Current smoker | 0.75 (0.44, 1.29) | 0.30 |
| Has an underlying condition | 0.44 (0.10, 1.88) | 0.27 |
| Received influenza vaccine this year | 0.78 (0.30, 2.02) | 0.61 |
| Household size (>3 people) | 0.34 (0.11, 1.05) | 0.06 |
| **BMI $\geq$ 30** | **1.86 (1.31, 2.63)** | **0.001** |
| Job involves client engagement | 0.65 (0.21, 2.00) | 0.46 |
| Length of employment (>12 months) | 1.61 (0.95, 2.73) | 0.08 |
| Formal health education | 1.02 (0.38, 2.78) | 0.96 |
| **Frequent[*] close contact[**] with clients** | **3.97 (1.06, 14.93)** | **0.04** |
| Frequent[*] direct physical contact[#] with clients | 1.64 (0.94, 2.86) | 0.08 |
| Frequent[*] close contact[**] with coworkers | 2.44 (0.98, 6.07) | 0.06 |
| Cleaning activities as part of job | 1.13 (0.27, 4.68) | 0.87 |
| Received training on COVID-19 cleaning | 1.55 (0.78, 3.11) | 0.22 |
| **Facility measures: safe distancing (>6 ft)** | **0.52 (0.32, 0.84)** | **0.01** |
| **Facility measures: masks for staff or clients** | **0.50 (0.30, 0.84)** | **0.01** |
| Any mask use for close contacts with clients | 1.40 (0.54, 3.61) | 0.49 |
| Always/mostly use gloves when in direct physical contact with clients | 0.82 (0.08, 8.47) | 0.87 |
| Use of mask when in close contact with coworkers | 1.46 (0.56, 3.82) | 0.44 |
| Always used masks when interacting with COVID-19 case | 1.44 (0.95, 2.22) | 0.09 |
| **Used gloves when interacting with COVID-19 case** | **3.90 (1.36, 11.19)** | **0.01** |
| **Thought about quitting because worried about COVID-19** | **1.84 (1.02, 3.33)** | **0.04** |

^n = 15 positive homeless shelter workers.

[+]$\alpha < 0.05$.

[*]Frequent defined as a few times a day to a few times a week.

[**]Close contact = within 6 feet for $\geq$15 minutes.

[#]Direct physical contact = touching.

increased risk of COVID-19 positivity, while facility mitigation measures of wearing masks or maintaining social distance was associated with decreased risk (PR 0.50, 95% CI 0.30, 0.84; p = 0.01; PR 0.52, 95% CI 0.32, 0.84; p = 0.01; respectively). Workers who thought about quitting their job because of concerns about COVID-19 had an increased risk of positivity (PR 1.84, 95% CI 1.02, 3.33; p = 0.04). We conducted a sensitivity analysis, excluding the 10 participants who said that they did not know their test results or who didn't answer the question (which was categorized as negative in main analysis); BMI, identifying as Non-Hispanic Black race, close contact with clients, social distancing, and wearing a mask remained statistically significant.

## Discussion

This study sought to characterize homeless shelter worker job practices, occupational exposures to SARS-CoV-2, and COVID-19 mitigation measures in the workplace. In this sample of homeless shelter workers, participants reported close contact and direct physical contact with

clients. Nearly 40% of workers reported having close contact with a person with known COVID-19 and all reported that they believed the contact occurred at work; 24% of those workers did not use masks all of the time during these interactions (Table 2). Workers who reported frequent contact with clients were more likely to test positive for SARS-CoV-2. Understanding how homeless shelter workers are exposed to COVID-19 at work is important to be able to implement mitigation strategies in this non-traditional workplace.

Recommendations for homeless service providers to help protect the staff and clients include hand hygiene and cleaning supplies, PPE (including masks), administrative controls (e.g., flexible work schedules), facility layout/ventilation considerations, and maintaining social distance [20]. All homeless shelter networks in our study had implemented at least one mitigation measure to reduce risk of worker exposure, with masks for staff/clients and handwashing being most common (Table 1). Most workers reported using masks most or all of the time when at work. We found that workers who reported that their facility implemented the mitigation measure of maintaining social distance and using masks were less likely to be infected (Table 4). Although staff should avoid handling client belongings [20], we found that about half of the workers reported touching client belongings or shared items. Workers often did not receive training in infection control or cleaning procedures. For example, 43% of workers had not received training on cleaning surfaces for SARS-CoV-2 even though it was a part of their job duties. Homeless shelter workers' training needs, both for their job tasks and for professional development, are often not prioritized [21]. Shelter management should fully implement recommendations and provide additional training on both COVID-19 and cleaning/disinfection [20].

In this study, workers reported being worried about becoming infected at work and had thought about quitting. Nearly 40% of workers had been at their shelter less than one year. Homeless shelter workers tend to be a transient work population with a high burnout rate [21–25]. Also, homeless shelter workers may be in a vulnerable population themselves since workers are sometimes residents who are offered jobs at the shelter [26]. Anecdotally, some homeless shelter workers in hourly wage unskilled jobs are, or have been, homeless before [27]. Homeless shelters are already prone to understaffing due to high turnover [21] which is magnified by COVID-19 isolation requirements if multiple workers test positive. COVID-19 has likely exacerbated the stress associated with working in homeless shelters.

Homeless shelter workers have many different job roles, including case managers, janitors, administrative/managerial staff, cooks, security guards and floor monitors. In our sample, close contact with clients was not limited to workers whose job duties included known client engagement; more than 50% of workers with administrative job duties also reported close or direct physical contact with clients. Thirty-six percent of homeless shelter workers said that they had close contact with a known COVID-19 case at work. Of the 15 who tested positive by non-blood testing, 80% believed that they were infected at work. Homeless shelter workers, regardless of primary job duties, may be at increased risk for COVID-19, due to frequent close contact in congregate settings [28]. Homeless shelter workers have been deemed essential critical infrastructure workers [5] thus would be recommended for COVID-19 vaccination in Phase 1B (i.e., frontline essential workers) [29].

Several demographic factors have been shown to increase risk of SARS CoV-2 infection in the general population, including older age, race, ethnicity, and obesity [30]. In this survey, almost 50% of participants were people of color whereas Whites make up the majority of the labor force (77% in 2019) [31]. Additionally, 36.8% of respondents were obese, which was slightly higher than obesity among U.S. employed adults (32.5% in 2016) [32]. We found that respondents who identified as Non-Hispanic Black or were obese were more likely to test positive. People of color are more likely to be employed in occupations with close proximity to

others [33]. People with higher risk factors for SARS CoV-2 infection may be over-represented in the homeless shelter workforce when compared to the general U.S. labor force.

During the survey period, many homeless shelters in major cities were conducting serial testing for SARS-CoV-2 of clients and staff [1]. In this study, 61% of participants were tested at the workplace and 69% did not experience symptoms around the time of testing, which supports the premise that many of these workers were tested as part of a universal screening process rather than because they were symptomatic. Universal screening could also explain why some workers were tested multiple times (range 1–10 tests). Over 60% continued to work while waiting for test results. It is not clear whether workers who tested positive would have been detected based solely upon symptoms. Of the 21 positive non-blood testing events, 76% were symptomatic around the time of testing, 6.3% worked while symptomatic and 63% sought medical care for their symptoms (Table 3). Serial testing at workplaces has been recommended as a control method to interrupt transmission of COVID-19 in possible outbreak situations [34].

This study is subject to several limitations. The questionnaire was online and self-administered where there may have been selection bias (e.g., access to internet, access to survey during working hours) and recall bias when reporting exposures, symptoms, and timing of testing. There were small number of respondents who reported positive tests which limited our ability to conduct more robust analyses to evaluate potential occupational risk factors associated with testing positive (Table 4). The respondents were a self-selected convenience sample where response rate was not able to be estimated. It is possible that workers who were more concerned with COVID-19 participated in the survey. Access to internet and ability to access survey during working hours may have been a factor as to who was able to participate. In addition, homeless shelter workers from multi-facility networks in large cities were invited to participate; thus, this sample may not be representative of all homeless shelter workers in the United States.

This is the first study that describes the work environment of homeless shelter workers in the context of COVID-19. We found that surveyed homeless shelter workers reported frequent close contact with clients; this was associated with having a positive test for SARS-CoV-2, while wearing masks and maintaining social distance at work were protective. Shelter management should continue to follow public health recommendations [20] by reinforcing mitigation measures in the workplace and training staff routinely on mitigation measures. Homeless shelter workers are essential workers with the potential for high-risk exposures, including close and direct physical contact with clients. Further research is needed on describing work environment, COVID-19 risks, and mitigation measures to reduce risk of infections among homeless shelter staff.

## Supporting information

**S1 Data. Plos one data dictionary.**
(PDF)

**S1 Dataset. Plos one dataset.**
(XLSX)

**S1 File. Homeless shelter worker survey.**
(PDF)

## Acknowledgments

Meagan Kay, Matthew Hanson, Margaret D. Luckoff, Jody Rauch, Libby Page, Public Health-Seattle & King County; Angela McCauley, Baltimore City Mayor's Office of Homeless Services; Julie L. Self, Centers for Disease Control and Prevention; Jessie Gaeta, Boston Health Care for the Homeless Program; Gerry Thomas, Boston Public Health Commission; Kate Tettamant, Georgia Department of Community Affairs; Tair Kiphibane, Salt Lake County Health Department.

**Disclaimer**: The findings and conclusions in this report are those of the authors and do not necessarily represent the official position of the Centers for Disease Control and Prevention/ the Agency for Toxic Substances and Disease Registry.

## Author Contributions

**Conceptualization:** Carol Y. Rao, Karin Huster.

**Data curation:** Carol Y. Rao, Tashina Robinson, Ryan Keating.

**Formal analysis:** Carol Y. Rao, Tashina Robinson.

**Investigation:** Carol Y. Rao, Farrell A. Tobolowsky, Emily Mosites.

**Methodology:** Karin Huster, Emily Mosites.

**Project administration:** Carol Y. Rao, Rebecca L. Laws, Farrell A. Tobolowsky, Temet M. McMichael, Elysia Gonzales, Emily Mosites.

**Resources:** Emily Mosites.

**Supervision:** Carol Y. Rao, Rebecca L. Laws, Elysia Gonzales, Emily Mosites.

**Validation:** Ryan Keating, Farrell A. Tobolowsky, Temet M. McMichael.

**Writing – original draft:** Carol Y. Rao, Rebecca L. Laws, Emily Mosites.

**Writing – review & editing:** Carol Y. Rao, Tashina Robinson, Karin Huster, Rebecca L. Laws, Ryan Keating, Farrell A. Tobolowsky, Temet M. McMichael, Elysia Gonzales, Emily Mosites.

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
