## [Decision Letter · Decision Letter 0]

29 Jul 2021

PONE-D-21-22300

Occupational exposures and mitigation strategies among homeless shelter workers at risk of COVID-19

PLOS ONE

Dear Dr. Rao,

Thank you for submitting your manuscript to PLOS ONE. After careful consideration, we feel that it has merit but does not fully meet PLOS ONE’s publication criteria as it currently stands. Therefore, we invite you to submit a revised version of the manuscript that addresses the points raised during the review process.

Both reviewers recommend minor revisions of text.

We look forward to receiving your revised manuscript.

Kind regards,

David M. Ojcius

Academic Editor

PLOS ONE

Journal Requirements:

2. Please provide additional details regarding participant consent. In the Methods section, please ensure that you have specified (1) whether consent was informed and (2) what type you obtained (for instance, written or verbal). If your study included minors, state whether you obtained consent from parents or guardians. If the need for consent was waived by the ethics committee, please include this information.

3. Please amend either the abstract on the online submission form (via Edit Submission) or the abstract in the manuscript so that they are identical

Reviewers' comments:

Reviewer's Responses to Questions

**Comments to the Author**

1. Is the manuscript technically sound, and do the data support the conclusions?

Reviewer #1: Yes

Reviewer #2: Yes

2. Has the statistical analysis been performed appropriately and rigorously? 

Reviewer #1: Yes

Reviewer #2: Yes

3. Have the authors made all data underlying the findings in their manuscript fully available?

Reviewer #1: Yes

Reviewer #2: No

4. Is the manuscript presented in an intelligible fashion and written in standard English?

Reviewer #1: No

Reviewer #2: Yes

5. Review Comments to the Author

Reviewer #1: In the introduction section – a comment should be made on the incidence of covid-19 infection in the general population versus that of homeless shelter workers.

Workers who reported “Don’t know” for a test result were categorized as a non-positive. This is very problematic. How can this be assumed? Are all workers informed of their test results? More justification should be given for why the study authors classified the data as such, if not this would make the data analysis very unreliable.

It is good that the limitation of convenience sampling is acknowledged. Are there reasons for the response rate of 63% (17 out of 27 shelters)? Are there specific characteristics of the 10 shelters that did not respond? Perhaps some information should be given on this.

Overall this is an interesting study, however there are big gaps in the study methodology which render the generalisability of the study’s findings questionable. It is good that the authors have acknowledged the limitations, however these limitations are pretty significant (In our sample, 15% of respondents reported testing positive compared to 4.3% of a universal testing database.). As such this study is of limited value.

There are some grammatical errors:

Line 275:

In our sample, close contact with clients was not limited to workers whose job duties with known client engagement;

Line 302:

There were small number of positives which limited our ability to conduct more robust analyses

Reviewer #2: This is a very well written manuscript that describes in detail the results of a multi-center homeless shelter workers survey. The survey is detailed and the data is robust related to respondents demographics, homeless shelter worker directed mitigation strategies and risk for covid infection. The tables are easy to read and understand.

The survey has touched on the mitigation strategies initiated in the center directed to shelter workers however does not go into details related to client related mitigation strategies and client education related to covid. If this information is available would be valuable.

I would recommend to expand a little more and emphasize in the discussion the section related to race/ethnicity and BMI and increased covid risk.

6. PLOS authors have the option to publish the peer review history of their article (what does this mean?). If published, this will include your full peer review and any attached files.

Reviewer #1: No

Reviewer #2: **Yes: **Dana Albon

---

## [Author Response · Author response to Decision Letter 0]

22 Sep 2021

Reviewer #1: 

a. In the introduction section – a comment should be made on the incidence of covid-19 infection in the general population versus that of homeless shelter workers,

RESPONSE: Thank you for this comment. The COVID-19 case data that is available for homeless shelter workers from https://nhchc.org/cdc-covid-dashboard/home/ is a cumulative count, rather than incidence. Also, the data are voluntarily submitted to the dashboard and are not a comprehensive count. CDC reports the cumulative incidence case rate (cumulative cases per 100,000) on a daily basis (CDC COVID Tracker https://covid.cdc.gov/covid-data-tracker/#trends_dailytrendscases). Available data for homeless shelter workers are prevalences only. Data on the cumulative incidence case rate of COVID-19 among homeless shelter workers are not available.

There is a recent publication, however, which is a meta-analysis of prevalences of COVID-19 among homeless populations, both in outbreak settings and non-outbreak settings. We have replaced the dashboard citation with this meta-analysis that is more robust than the NHCHC voluntary dashboard. (Mohsenpour A, Bozorgmehr K, Rohleder S, Stratil J, Costa D. SARS-Cov-2 prevalence, transmission, health-related outcomes and control strategies in homeless shelters: Systematic review and meta-analysis. EClinicalMedicine. 2021:101032.) We have revised the introduction (lines 61-68) in the marked up revised version of the manuscript.

b. Workers who reported “Don’t know” for a test result were categorized as a non-positive. This is very problematic. How can this be assumed? Are all workers informed of their test results? More justification should be given for why the study authors classified the data as such, if not this would make the data analysis very unreliable.

RESPONSE: We had conducted a sensitivity analysis that excluded the 10 respondents who reported “don’t know” for their testing result (N=67). The variables that were most important for work factors (i.e., close contact with residents and IPC measures) did not change between the two groups. In addition, race and obesity also did not change. The variables that did change (highlighted in yellow) only changed in significance level; magnitude and direction of the PRs did not change. We think this is a power issue with small sample sizes. The sensitivity analysis is described in lines 231-235 in the revised, marked up version of the manuscript.

In our experience with universal testing of homeless shelter workers, all workers were informed of their positive test results. Not all workers who tested negative, however, were informed of their negative results. Thus, we included the “don’t knows” as negatives in the modeling (N=77). Excluding the “don’t knows” would not change the conclusions or work-related recommendations of the study.

c. It is good that the limitation of convenience sampling is acknowledged. Are there reasons for the response rate of 63% (17 out of 27 shelters)? Are there specific characteristics of the 10 shelters that did not respond? Perhaps some information should be given on this.

RESPONSE: There seems to be some confusion over individual shelter (n=27) and shelter network (n=17). Some shelter networks had more than one individual shelter. There were 27 individual shelters within 17 shelter networks. All 27 individual shelters in the 17 shelter networks participated. For analysis, the 27 individual shelters that were managed by the same organization were grouped into their 17 shelter networks since management of staff and mitigation strategies would have been the same across the entire shelter network. We have tried to clarify this in the text in lines 121-122, 127-128, and 141 in the revised, marked up version of the manuscript.

d. Overall this is an interesting study, however there are big gaps in the study methodology which render the generalisability of the study’s findings questionable. It is good that the authors have acknowledged the limitations, however these limitations are pretty significant (In our sample, 15% of respondents reported testing positive compared to 4.3% of a universal testing database.). As such this study is of limited value.

RESPONSE: The 4.3% for the universal testing database is not directly comparable to the 15% prevalence among respondents. Study sites had to have had a positive case within their shelter to be included in the study versus anyone who can submit data to the universal testing database dashboard. Thus, it is not unexpected that the prevalence would be higher in this study population than the overall homeless shelter worker population since the respondents would have been at increased risk of exposure. In fact, when compared to the recently published meta-analysis (new citation Mohsenpour et al., 2021), the prevalence from our population (15%) was closer to the pooled prevalence among shelter workers during outbreaks (14.8%). The confusing sentence (line 324-325) regarding the universal testing database is deleted from the text.

e. There are some grammatical errors:

Line 275: In our sample, close contact with clients was not limited to workers whose job duties with known client engagement: Error fixed

f. Line 302: There were small number of positives which limited our ability to conduct more robust analyses: Error fixed

Reviewer #2: This is a very well written manuscript that describes in detail the results of a multi-center homeless shelter workers survey. The survey is detailed and the data is robust related to respondents demographics, homeless shelter worker directed mitigation strategies and risk for covid infection. The tables are easy to read and understand.

a. The survey has touched on the mitigation strategies initiated in the center directed to shelter workers however does not go into details related to client related mitigation strategies and client education related to covid. If this information is available would be valuable.

RESPONSE: The survey did not include questions on client related mitigation strategies or client education. There were questions on safe distancing and face coverings for staff/clients, which were included in Table 1. No additional information on mitigations strategies for clients is available for this survey.

b. I would recommend to expand a little more and emphasize in the discussion the section related to race/ethnicity and BMI and increased covid risk.

RESPONSE: Thank you for this important comment. Non-Caucasian race/ethnicity and BMI are known risk factors for COVID-19 infection and poor outcomes. Although the focus of the study is on modifiable work-related factors that could impact infection, we also agree that homeless shelter workers may have important demographic factors that could impact COVID-19 infections. The study population was 24.5% Non-Hispanic black and 36.8% with BMI > 30. A new paragraph to highlight these demographic factors has been added to lines 295-303 in the revised, marked up version of the manuscript.

---

## [Editor Report · Decision Letter 1]

30 Sep 2021

Occupational exposures and mitigation strategies among homeless shelter workers at risk of COVID-19

PONE-D-21-22300R1

Dear Dr. Rao,

We’re pleased to inform you that your manuscript has been judged scientifically suitable for publication and will be formally accepted for publication once it meets all outstanding technical requirements.

Kind regards,

David M. Ojcius

Academic Editor

PLOS ONE
---

## [Editor Report · Acceptance letter]

20 Oct 2021

PONE-D-21-22300R1 

Occupational exposures and mitigation strategies among homeless shelter workers at risk of COVID-19 

Dear Dr. Rao:

I'm pleased to inform you that your manuscript has been deemed suitable for publication in PLOS ONE. Congratulations! Your manuscript is now with our production department. 

Kind regards, 

on behalf of

Dr. David M. Ojcius 

Academic Editor

PLOS ONE